# A Integrated Dedicated Outdoor Air System to Optimize Energy Saving

**Yew Khoy Chuah * and Jun Jie Yang**

Department of Energy and Refrigerating Air-conditioning Engineering, National Taipei University of Technology, Taipei 10608, Taiwan; k19800575@gmail.com
* Correspondence: yhtsai@ntut.edu.tw; Tel.: +886-2-27712171; Fax: +886-2-87733713

**Abstract:** Outdoor air supply is required to maintain good indoor air quality (IAQ). For tropical or subtropical regions, warm and humid outdoor air would cause excess air-conditioning energy use. This study has proposed an integrated dedicated outdoor air system (IDOAS), which integrates the enthalpy exchange and outdoor air cooling into a unitary system. IDOAS could operate independently of central air-conditioning systems thus saving tremendous piping cost and energy needed to deliver chilled water to outdoor air unit in a conventional centralized system. An experimental unit of IDOAS was built to prove this novel concept. Enthalpy exchange efficiency was tested to be about 44%. The test results show that about 44% of energy needed to condition the outdoor air can be saved. A reverse Rankine refrigeration cycle was integrated to cool the outdoor air. Due to this integrated configuration, the air passing through the condenser would be at a lower temperature. The consequent lower refrigerant condensing temperature would improve the cooling cycle efficiency. The cooling coefficient of performance (COP) was improved by about 46%. In addition, the outdoor air could be conditioned to a lower humidity before being supplied to space, which would improve the thermal comfort. The test results of this novel IDOAS show that it could provide good air quality at lower energy use.

**Keywords:** outdoor air; IAQ; energy saving; enthalpy heat exchange; green building

## 1. Introduction

Indoor air quality has become an indoor environment problem worldwide. The Indoor Air Quality Act of Taiwan [1] was enacted in 2012 and it became one of the few countries which imposed a regulation on indoor air quality. The indoor air quality standards imposed by the Indoor Air Quality Act of Taiwan are given in Table 1. Public health was the major concern in the enactment of this regulation. There are nine items of air quality standards. In Table 1, 8 h means 8 h average value, so for 1 h and 24 h. The standards were first imposed in public spaces such as public transport stations, hospitals, and later all levels of education institutions. Spengler and Sexton [2] discussed the risk of indoor air pollution with a public health perspective. They mentioned that most people stay indoors about ninety percent of the time and the exposure to indoor pollution was a health issue.

**Table 1.** Taiwan indoor air quality standards.

| Item | Standard | | Unit |
|:---:|:---|:---:|:---:|
| $CO_2$ | 8 h | 1000 | ppm |
| CO | 8 h | 9 | ppm |
| HCHO | 1 h | 0.08 | ppm |
| TVOC | 1 h | 0.56 | ppm |
| Bacteria | Maximum | 1500 | $CFU/m^3$ |

**Table 1.** *Cont.*

| Item | | Standard | Unit |
|---|---|---|---|
| Fungi | Maximum | 1000 or ratio of indoor to outdoor ≤1.30 | $CFU/m^3$ |
| $PM_{10}$ | 24 h | 75 | $\mu g/m^3$ |
| $PM_{2.5}$ | 24 h | 35 | $\mu g/m^3$ |
| $O_3$ | 8 h | 0.06 | ppm |

Monitoring all the items in Table 1 would be very costly. In the guiding principles, only items identified with potential sources are needed to be monitored. The selection is also based on the risk assessment of public health. In general, fresh air ventilation is a mean to maintain good air quality. It is often said a solution to pollution is dilution.

South Korea had also imposed regulations on indoor air quality. The revised Act in 2003 [2] increased the number of pollutant standards. In some countries or regions, the IAQ Certification Scheme is promoted with the same purpose to improve indoor air quality, such as that carried by the Hong Kong government [3].

It is generally known that fresh outdoor air is important to maintain good indoor air quality (IAQ). Early in 1992, Levenhagen [4] proposed using a dedicated outdoor air system (DOAS) to ensure a proper ventilation rate. Mumma [5] in 2001 proposed a design of DOAS. Other related references suggested using DOAS to handle air-conditioning latent load [6–9]. Mumma [10] gave an overview of integrating dedicated outdoor air systems with parallel terminal systems. Shank and Mumma [11] further proposed using DOAS to handle the latent cooling load, working with distributive parallel systems to handle sensible cooling. Chien et al. [12] presented experimental results of radiant floor cooling. They proposed that a parallel system such as DOAS be used to handle latent load. Li et al. [13] considered a multistage cooling design for DOAS for the purpose of supplying lower air temperature.

DOAS with heat recovery was considered in some research [14,15]. Chuah and Lin [14] analyzed the air-conditioning load for a typical office building in Taiwan. The results showed that about 27% of air-conditioning energy was used to condition the outdoor air to the indoor conditions. Lower $CO_2$ concentration means more outdoor air has to be supplied to the space. Then, more energy would be needed to cool and dehumidify the outdoor air. James [16] also mentioned that a significant portion of commercial air-conditioning load was due to outdoor air.

Currently, the common practice in central air-conditioning systems is by using an outdoor air handling unit (OAU) to bring in sufficient outdoor air. Outdoor air will be cooled to indoor conditions of temperature and humidity. Such practices have the disadvantage of coupling to central air-conditioning system operation. In such practices, pumping energy to transport chilled water from the central plant to OAU is costly. As shown in Figure 1, chilled water from the central system has to be pumped to the OAU via piping. Normally, the piping construction cost is quite high. The indoor air quality standard of ASHRAE in 2013 [17] further required for fresh air to be effectively distributed to all parts of a building. ASHRAE later published a technical guide for designing DOAS [18]. The above discussion indicates that conventional OAU has the disadvantage of coupling to the central system. The current outdoor air unit (OAU) is generally designed with the purpose to ensure proper outdoor air supply, often not considering the piping construction cost and energy cost of OAU. As seen in Figure 1, OAU is normally designed to distribute outdoor air to each space that would further cause ducting cost and fan energy. Due to climate change, warmer outdoor air could only result in more energy use by an air-conditioning system, especially that to condition the outdoor air.

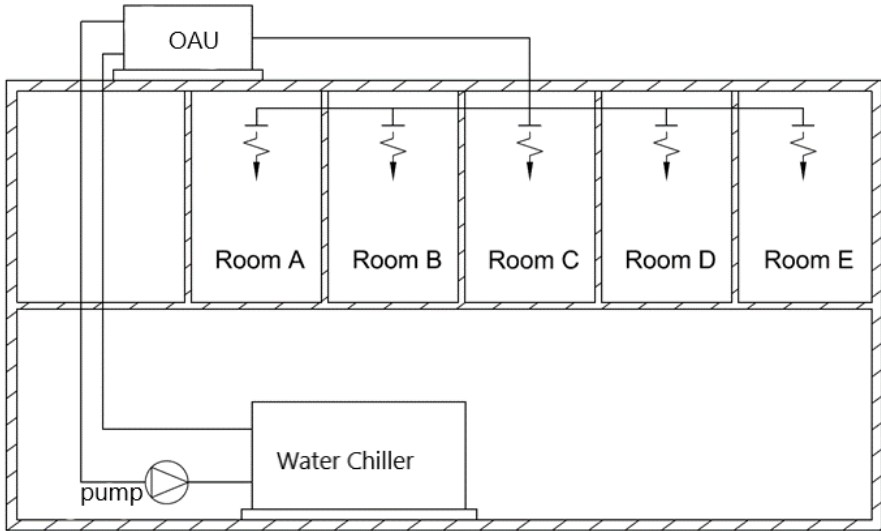

**Figure 1.** Outdoor air supply unit of central air-conditioning system.

Mui et al. [19] attempted to analyze the air-conditioning energy use in some Hong Kong office buildings. They concluded the energy use was strongly related to the $CO_2$ concentration. $CO_2$ concentration is strongly related to the outdoor air supply rate. This further shows that more fresh air ventilation would result in higher air-conditioning energy use.

Chaeand and Strand [20] attempted using computer modeling to understand the energy performance of DOAS. Mumma [21] further studied the feasibility of using DOAS to maintain a building at a positive pressure preventing excess leakage of outdoor air. Xiao et al. [22] applied a liquid desiccant method to dehumidify the outdoor air but the liquid desiccant method needed cooling to maintain the liquid temperature. There were also research studies on applying solid desiccant to remove moisture from fresh air such as that discussed by Liu et al. [23]. Either using liquid or a solid desiccant would have to obey the thermodynamic law. The heat of dehumidification has to be removed by some means of cooling. Therefore, the advantage of desiccants would be constrained.

The review of the above literature indicates the problem of reducing energy use while maintaining good IAQ. Global warming effects are becoming more eminent as days pass. This study was then an attempt to investigate a mean of saving air-conditioning energy while maintaining good indoor air quality. A novel integrated unitary outdoor air system (IDOAS) was studied, an experimental unit was fabricated, and then tested in this study.

## 2. The Advantage of Integrated Unitary Dedicated Ventilation Unit

Due to the large enthalpy difference of indoor and outdoor air, it would cause excessive air-conditioning energy use to cool and dehumidify the outdoor air to room air conditions. Enthalpy exchange between room exhaust air and outdoor air intake is generally known to be effective in reducing the enthalpy (energy state) of outdoor air. This study thus proposed a novel design that integrated enthalpy exchange and air cooling into a unitary system. Such a design has the advantage of distributive unitary systems instead of centralized outdoor air supply. A unitary system can be individually installed and decoupled from the central system. Therefore, tremendous cost of piping, ducting not the least fan, and pump energy can be saved. As mentioned, and explained in Figure 1, a conventional centralized outdoor air unit (OAU) would be coupled with the operation of a central air-conditioning system and cannot operate independently when fresh air is required but not cooling.

The ventilation unit proposed was a unitary integrated dedicated outdoor air system as shown in Figure 2, hereafter referred to as IDOAS. It can be seen in Figure 2 that an IDOAS consists of an enthalpy exchange unit and a refrigeration cycle cooling unit in a unique configuration. This configuration is important so that optimal energy saving can be achieved. It is shown in Figure 2a that the refrigerant

compressor, refrigerant tubes, expansion valve, condenser, and evaporator are all housed in a unit with a total heat exchanger (enthalpy exchanger).

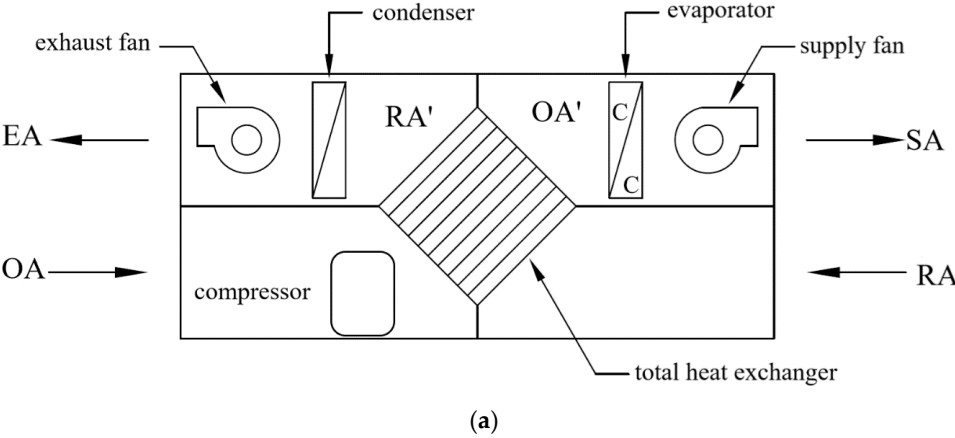

(**a**)

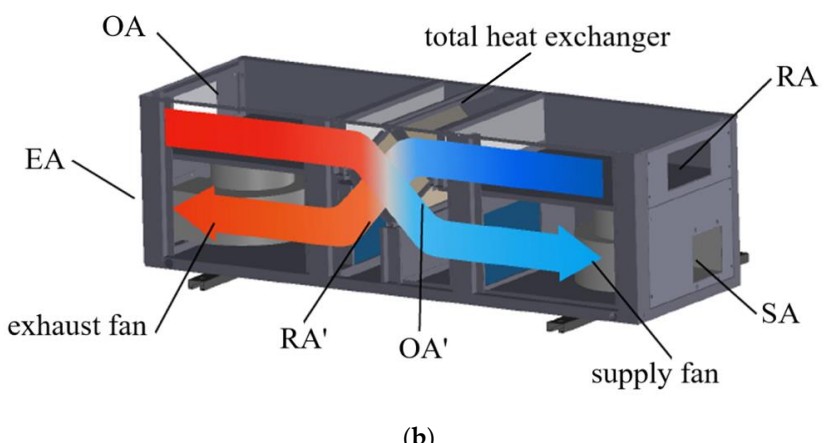

(**b**)

**Figure 2.** The configuration of an integrated dedicated outdoor air system (IDOAS). (**a**) The major components of IDOAS. (**b**) The air flows and heat exchange through the total heat exchanger.

Figure 2b shows that room air (RA) would undergo total heat exchange with outdoor air (OA) and becomes RA′. RA′ will then flow through the refrigerant condenser before being exhausted as exhaust air (EA). With this configuration, the condenser would be cooled by a lower temperature RA′ (compared to OA for an air-cooled system). Consequently, the system will have a lower refrigerant condensing temperature. In terms of the laws of thermodynamics, the refrigerant cycle efficiency will be higher at a lower refrigerant condensing temperature. This explains the basics that IDOAS could operate at a higher refrigeration cycle efficiency.

It is also noted in Figure 2 that OA becomes OA′ at a cooler and drier state after total heat exchange with RA. OA′ would be further cooled by a refrigerant evaporator and passing out as supply air (SA). In the process heat, and moisture will be further removed from OA′. Therefore, the outdoor supply air (SA) will be at a state of lower temperature and humidity. Temperature and humidity are important factors of thermal comfort. This explains that IDOAS could provide outdoor air at states that would enhance the thermal comfort for the air-conditioned space. Besides the advantages as mentioned, IDOAS can be installed at the point of use as unitary systems, thus can be more effectively distribute outdoor air to air-conditioned spaces.

A model unit of IDOAS was built for experimental tests. A picture of the test unit of IDOAS is shown in Figure 3. Figure 3 shows that IDOAS has all components housed in a single unit with the configuration shown in Figure 2. The refrigerant condenser and the evaporator are integrated into IDOAS. The refrigerant condenser is air-cooled by design but not cooled by the outdoor air,

and instead, it will be cooled by exhaust room air at a much lower temperature. As for the evaporator, the refrigerant undergoes direct expansion, absorbs heat, and removes moisture from OA'. On the contrary, for OAU, as shown in Figure 1, chilled water (say at 7 °C) has to be pumped to the OAU to cool the air.

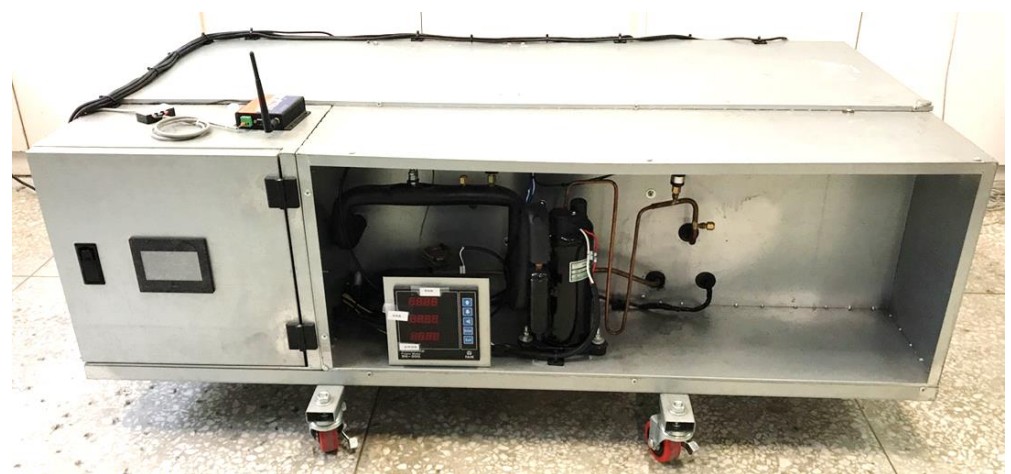

**Figure 3.** The IDOAS fabricated.

The energy saving principles of IDOAS can be explained using a psychrometric chart as shown in Figure 4. The ordinate of the psychrometric chart is the humidity ratio given as kg vapor in a kg of dry air (or g vapor/kg dry air), often referred to as the absolute humidity of air. The abscissa is the air temperature. Enthalpy is the slant axis on the psychrometric chart with the unit as kJ/kg of dry air. In the psychrometric chart, outdoor air (OA) after the total heat exchanger is brought to OA' with much lower enthalpy. RA' after the total heat exchanger will be at higher enthalpy. The increase of enthalpy for RA in principle will be equal to that of the enthalpy decrease of OA. Therefore, total exchange is often referred to as enthalpy exchange. In air-conditioning, the energy state of air is evaluated as enthalpy as indicated in Equation (1). In Equation (1), temperature $T$ is evaluated as °C and enthalpy $h$ at kJ/kg.

$$h = c_{pa}T + \omega\left(c_{pw}T + h_g\right) \tag{1}$$

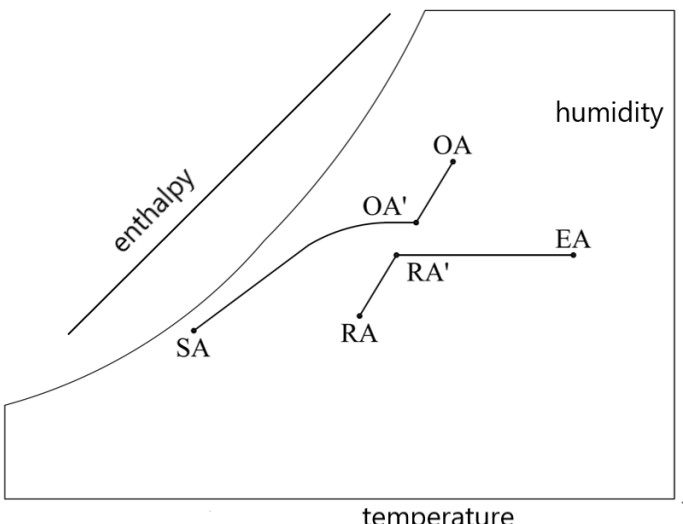

**Figure 4.** Psychrometrics of IDOAS.

In Equation (1), $\omega$ is the humidity ratio (kg vapor/kg dry air), $c_{pa}$ and $c_{pw}$ are, respectively, the specific heats of dry air and water vapor; $h_g$ is the water vapor enthalpy at 0 °C.

A cross-flow-type total heat exchanger was used in this study. A typical configuration of a cross-flow type total heat exchange is shown in Figure 5. The cross-flow of two air streams flow through alternate permeable layers of the total heat exchanger. Total heat exchange between OA and RA would reduce the energy state change of OA to OA'. For a total heat exchanger, the corrugated structure is impregnated with some kind of desiccant, for example silica gel. The humidity ratio of OA is higher than that of the equilibrium value at the desiccant interface. The humidity ratio represents the concentration of moisture in air. In principle, the moisture in OA will be adsorbed by the desiccant. The adsorbed moisture would then permeate to the other side of the permeable layers. RA has a lower humidity ratio than the equilibrium value at the desiccant surface. Therefore, moisture will be desorbed from the desiccant into the RA air stream. This explains the latent heat exchange. The sensible exchange is by the principle of the heat exchange of air streams between the alternate layers. The above explains the principles of total heat exchange (sensible plus latent) or the enthalpy exchange that designed into the IDOAS.

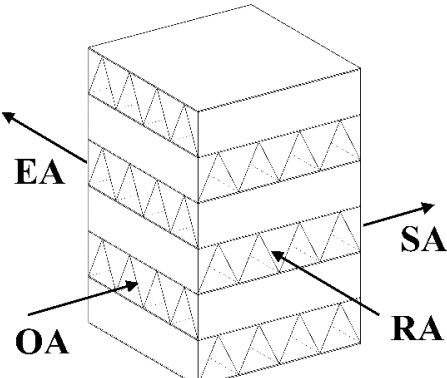

**Figure 5.** A total heat exchanger.

Taiwan is situated in the subtropical region located about 23 degrees north in latitude. During the summer months in Taiwan, outdoor air (OA) temperature is often higher than 35 °C, especially in the afternoon. Air-cooled air-conditioners are generally rated at low energy efficiency. This is due to the condenser refrigerant cooled by the high ambient temperature. The air temperature is the key factor for energy efficiency. For IDOAS, as indicated in Figures 2 and 4, a lower temperature of RA' instead is used to cool the refrigerant in the condenser. The energy state change from RA' to EA is the cooling of the refrigerant in the condenser. The energy state change from OA' to SA represents the removal of heat and moisture from OA'. This explains the advantages of this novel ventilation unit, which can provide sufficient high-quality fresh air that is low in energy consumption.

## 3. Experimental Tests

The measurements in the experimental unit are shown in Figure 6. The measurements included the air flow rates separately for exhaust room air and outdoor air intake. The calculation of total heat exchange efficiency requires the measurements of temperature and humidity (T, RH) before and after the total heat exchanger for the two air streams. Enthalpy change for the two air streams (room exhaust and outdoor air) were calculated from temperature and humidity measurements as indicated in Equation (1). Theoretically, enthalpy exchange efficiency $\varepsilon$ can be calculated from the enthalpy change of *OA* or the enthalpy change of *RA* as shown in Equation (2). The calculation of enthalpy exchange efficiency is based on the theoretical maximum enthalpy exchange from *OA* to *RA* $\dot{m}_{min}(h_{OA} - h_{RA})$.

$$\varepsilon = \frac{\dot{m}_{OA}(h_{OA} - h_{OA'})}{\dot{m}_{min}(h_{OA} - h_{RA})} = \frac{\dot{m}_{RA}(h_{RA'} - h_{RA})}{\dot{m}_{min}(h_{OA} - h_{RA})} \tag{2}$$

The air flow rates were calculated as the average velocity multiplied by the flow areas of IDOAS. Three velocities (test point v in Figure 6) were measured for each flow area to calculate the average velocity. The power consumption of the compressor was measured by a multi-function power meter shown in Table 2. Temperature, humidity, and velocity probes were connected to the testo multi-function analyzer to measure and collect the measurement data.

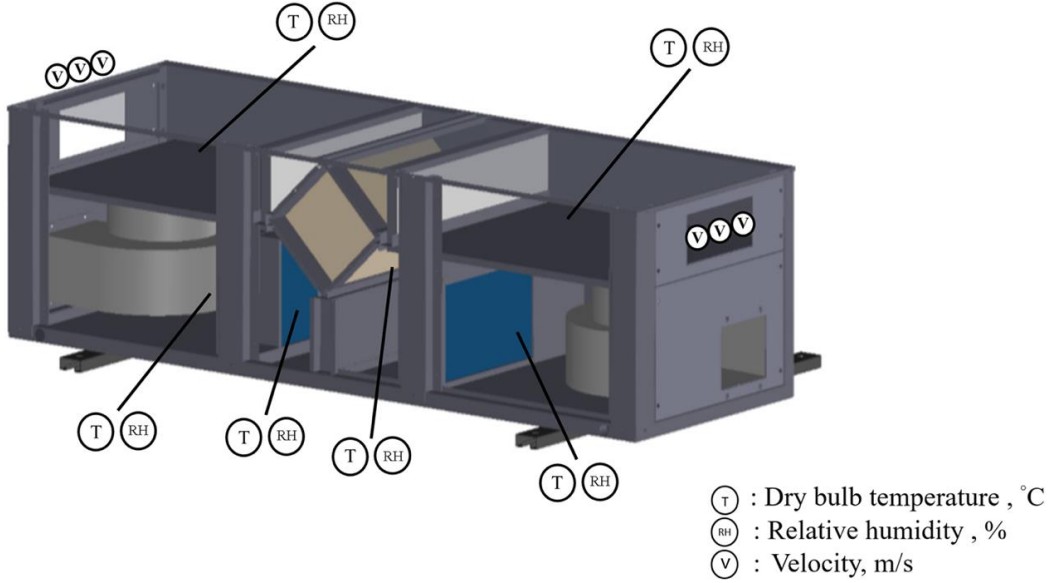

: Dry bulb temperature , °C
: Relative humidity , %
: Velocity, m/s

**Figure 6.** The measurements in the experimental tests.

**Table 2.** Instruments used in the experiments after calibration.

| Instrument | Made | Specifications and Accuracy |
| --- | --- | --- |
| Multi-function analyzer | testo 480 | Probes made by testo can be connected to 480 for measuring temperature, humidity, and air velocity |
| Vane velocity probe | testo | Diameter 60 mm, range from 0.25 to 20 m/s, accuracy ±0.1 m/s |
| Temperature probe | testo | Range from −20 to +70 °C, accuracy ±0.1 °C |
| Humidity probe | testo | Range from 0% to 100% RH, ceramic capacity type, accuracy 2% RH |
| Multi-function power meter | TAIK S6–300 | Measure voltage, current, and power, accuracy ±0.2% |

The cooling rate (W) of IDOAS was the air cooling rate by the evaporator calculated using Equation (3).

$$\text{Cooling rate } \dot{Q} = \frac{1}{3}\sum_{1}^{3}\rho v_i \; x \left[ A \; x \; (h_i - h_{o)} \right] \tag{3}$$

The first term $\frac{1}{3}\sum_{1}^{3}\rho v_i$ is the mass flux (kg/(s.m$^{-2}$)) and $v_i$ is the velocities measured. $h_i$ and $h_o$ are the enthalpy before and after flowing through the evaporator. The energy efficiency of the cooling cycle is represented by the coefficient of performance (COP), which is calculated using Equation (4).

$$\text{COP} = \frac{\dot{Q}}{W} \tag{4}$$

In Equation (4), $\dot{Q}$ is the cooling rate as calculated using Equation (3) and W is the power measured for the refrigerant compressor. The instruments used to measure the performance of IDOAS are described in Table 2. Power use of the refrigerant compressor was measured by a multi-function power meter.

A small experimental test unit of IDOAS was designed, fabricated, and tested. The specifications of the experimental test unit are described in Table 3. The experimental unit was designed with a ventilation rate of about 450 m³/h, sufficient for a space of 15 persons [15]. The compressor was supplied by a Taiwanese compressor manufacturer. The total heat exchanger was made in Japan.

The refrigerant condenser and the evaporator were sized by the authors and supplied by a local coil company. The objectives of the experiments were to prove the novelty of IDOAS, to understand the effectiveness of enthalpy exchange, and the refrigerant cycle cooling performance. The energy saving potential of IDOAS was also evaluated.

**Table 3.** The experimental IDOAS unit.

| Item | Specifications |
|---|---|
| Compressor | Constant speed type, R410A refrigerant, rated cooling capacity 1900 W, and COP 2.90 |
| Total heat exchanger | 300 mm × 175 mm, cross-flow type |
| Condenser | 3423 W at 35 °C air temperature |
| Evaporator | 2867 W at 26 °C air temperature |
| Air volume | 450 m³/h or 0.125 m³/s |

## 4. Test Results and Discussion

The IDOAS had two variable speed controlled fans. Several tests were carried out to understand the fan performance. The designed supply air volume was 0.125 m³/s. The test results of fan frequency and the corresponding air volume are given in Table 4. Graphical plots of frequency with air flow volume are given in Figure 7. The two fans were connected to a single power source thus operated at the same frequency. The frequency chosen was 1280 Hz, and the flow volumes at this frequency were 0.132 m³/s and 0.1206 m³/s, respectively, for the exhaust and supply fans.

**Table 4.** Tested air volume at different frequency.

| Frequency, Hz | Exhaust Fan | | Supply Fan | |
|---|---|---|---|---|
| | Air Volume (m³/s) | Power (kW) | Air Volume (m³/s) | Power (kW) |
| 1600 | 0.1576 | 0.118 | 0.1749 | 0.108 |
| 1280 | 0.1137 | 0.062 | 0.1288 | 0.058 |
| 960 | 0.0830 | 0.030 | 0.0860 | 0.026 |

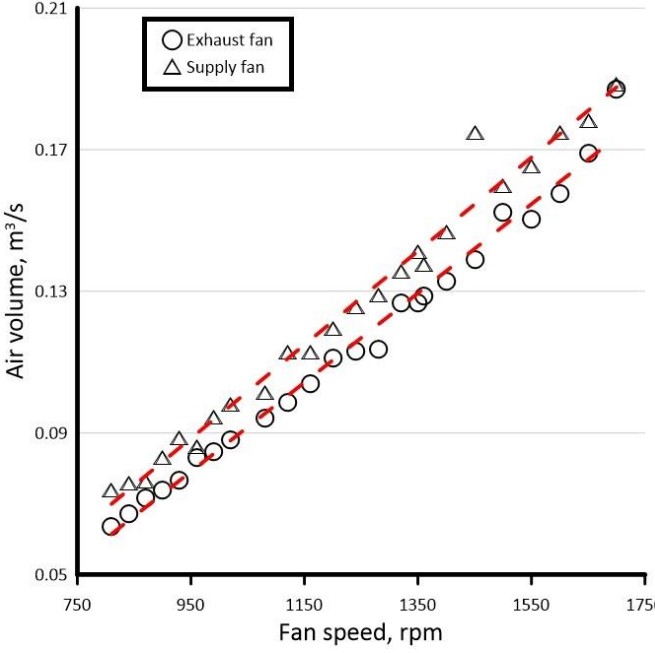

**Figure 7.** Graphical plots of frequency with air flow volume for fans.

## 5. Total Heat Exchange Performance

The test results of total heat exchanger performance are given in Tables 5 and 6. In Table 5, it is noted that OA′ was about 6 °C lower than OA. The humidity ratio of OA′ was lower than that of OA by about 2.29 g/kg. More significantly, enthalpy of OA′ was lower than OA by 15.33 kJ/kg. It would mean that 15.33 kJ of cooling can be saved per kg of outdoor air. The reduction of about 2.29 g/kg of moisture from OA is also significant. It shows that OA had been cooled and at the same time dehumidified without mechanical means but just through the total heat exchange with the room exhaust air. The cooler and drier air after flowing through the evaporator will be conditioned to an even lower temperature and lower humidity supply air (SA) would, in effect, improve the air quality, and at the same time increase the thermal comfort.

**Table 5.** The total heat exchange for outdoor air.

| OA States | | | | OA′ States | | | |
|---|---|---|---|---|---|---|---|
| Temp, °C | RH% | Humidity ratio g/kg | h, kJ/kg | Temp, °C | RH% | Humidity ratio g/kg | h, kJ/kg |
| 33.1 | 57.8 | 18.57 | 81.97 | 27.2 | 72.2 | 16.49 | 69.41 |
| 33.0 | 58.7 | 18.77 | 81.25 | 27.1 | 72.0 | 16.34 | 68.93 |
| 33.2 | 58.2 | 18.82 | 81.59 | 27.3 | 71.7 | 16.47 | 69.46 |
| Average OA enthalpy = 81.60 kJ/kg | | | | Average OA′ enthalpy = 66.27 kJ/kg | | | |
| Average OA humidity ratio = 18.72 g/kg | | | | Average OA′ humidity ratio = 16.43 g/kg | | | |

**Table 6.** The total heat exchange for exhaust room air.

| RA States | | | | RA′ States | | | |
|---|---|---|---|---|---|---|---|
| Temp, °C | RH% | Humidity ratio g/kg | h, kJ/kg | Temp °C | RH% | Humidity ratio g/kg | h, kJ/kg |
| 21.5 | 59.5 | 9.55 | 45.87 | 29.7 | 44.3 | 11.61 | 57.88 |
| 21.6 | 59.3 | 9.58 | 46.04 | 28.9 | 45.5 | 11.38 | 57.73 |
| 21.4 | 60.0 | 9.57 | 45.73 | 28.5 | 46.6 | 11.39 | 57.86 |
| Average RA enthalpy = 45.88 kJ/kg | | | | Average RA′ enthalpy = 57.82 kJ/kg | | | |
| Average RA humidity ratio = 9.57 g/kg | | | | Average RA′ humidity ratio = 11.46 g/kg | | | |

Similar analyses of the test results are given for the exhaust indoor room air (RA) as shown in Table 6. It is noted in Table 6 that an enthalpy exchange of about 11.94 kJ/kg was obtained, closely in agreement with the enthalpy exchange of outdoor air. This congruence in energy balance indicates the reliability of the test results.

## 6. Energy and Efficiency Analysis

The heat exchange efficiency was calculated using the average values in Tables 5 and 6. The total heat exchange efficiency was calculated based on enthalpy, which was calculated from the measurements of temperature and humidity using Equation (1). Equations (5) and (6) were used to calculate the enthalpy exchange efficiency, respectively, for OA and RA.

$$\varepsilon_{OA \rightarrow OA'} = \frac{\Delta h}{\Delta h_{max}} = \frac{\dot{m}_{OA}(h_{OA} - h_{OA'})}{\dot{m}_{min}(h_{OA} - h_{RA})} = 37.82\% \tag{5}$$

$$\varepsilon_{RA \rightarrow RA'} = \frac{\Delta h}{\Delta h_{max}} = \frac{\dot{m}_{RA}(h_{RA'} - h_{RA})}{\dot{m}_{min}(h_{OA} - h_{RA})} = 33.43\% \tag{6}$$

The calculated results are also shown in Equations (5) and (6), which indicate again the accuracy of the measurements. It is to be noted here that total heat exchange (enthalpy exchange) efficiency is related to the dimensions. The experimental unit was of small size and the total heat exchange efficiency tested agrees well with the supplier's data.

The heat exchange efficiency could be different for sensible heat and latent heat. Sensible means effect of temperature change and latent means humidity change. The sensible heat exchange efficiency for OA and RA were calculated and given, respectively, in Equations (7) and (8). It is noted that sensible heat exchange is higher than total heat exchange as given in Equations (5) and (6).

$$\varepsilon_{SH,OA\to OA'} = \frac{\Delta T}{\Delta T_{max}} = \frac{\dot{m}_{OA}(T_{OA} - T_{OA'})}{\dot{m}_{min}(T_{OA} - T_{RA})} = 55.7\% \tag{7}$$

$$\varepsilon_{SH,RA\to RA'} = \frac{\Delta T}{\Delta T_{max}} = \frac{\dot{m}_{RA}(T_{RA'} - T_{RA})}{\dot{m}_{min}(T_{OA} - T_{RA})} = 64.94\% \tag{8}$$

The latent heat exchange efficiency was also computed from the test results, given, respectively, for OA and RA in Equations (9) and (10). It is noted that latent exchange efficiencies of 27.39% and 20.68% are much lower than that of sensible heat exchange.

$$\varepsilon_{LH,OA\to OA'} = \frac{\Delta \omega}{\Delta \omega_{max}} = \frac{\dot{m}_{OA}(\omega_{OA} - \omega_{OA'})}{\dot{m}_{min}(\omega_{OA} - \omega_{RA})} = 27.39\% \tag{9}$$

$$\varepsilon_{LH,RA\to RA'} = \frac{\Delta \omega}{\Delta \omega_{max}} = \frac{\dot{m}_{RA}(\omega_{RA'} - \omega_{RA})}{\dot{m}_{min}(\omega_{OA} - \omega_{RA})} = 20.68\% \tag{10}$$

Latent heat exchange is important to regions of high humidity. Higher latent heat exchange efficiency would be beneficial to regions with high humidity. Global warming has resulted in warmer summers in high latitude countries, where sensible heat exchange could be more important.

The cooling and heat rejection rate of IDOAS were also measured. The cooling rate is the heat removal at the evaporator $Q_{evap}$, and the heat rejection rate $Q_{cond}$ is the heat rejected by the cooling unit. $Q_{evap}$ and $Q_{cond}$ were measured to be 2.454 kW and 3.361 kW given, respectively, in Equations (11) and (12). The calculations were based on OA′ and RA′, at states after the total heat exchanger. $Q_{evap}$ is the cooling rate required to condition the air to the supply conditions SA.

$$Q_{evap} = \dot{m}_{OA}(h_{OA'} - h_{SA}) = 2.454 \, kW \tag{11}$$

$$Q_{cond} = \dot{m}_{RA}(h_{EA} - h_{RA'}) = 3.361 \, kW \tag{12}$$

Due to the effects of total heat exchange, the cooling capacity can be significantly reduced. The result shown in Equation (13) was calculated using the average values of OA and OA′ in Table 5. Energy saving is quite significant, otherwise $Q_{evap}$ would be 4.41 kW instead (2.454 + 1.956), a saving of 44.3% (1.956/4.410).

$$Q_{THX,OA\to OA'} = \dot{m}_{OA}(h_{OA} - h_{OA'}) = 1.956 \, kW \tag{13}$$

The standard rating of outdoor air temperature for the performance test of air-conditioners for cooling is 35 °C [24]. For IDOAS, the air temperature entering the condenser was about 29 °C in this study, significantly lower than 35 °C. This 6 °C temperature difference would reduce the condensing temperature and increase cooling efficiency.

Isentropic efficiency of the compressor was measured to be 0.78, obtained by comparing the compressor power to that of isentropic compression. The compressor was of constant speed with the rated COP at 2.90. The coefficient of performance COP is defined as in Equation (3), calculated as the ratio of cooling rate against the power input into the compressor. In this study, COP was measured to be 4.25. Therefore, the increase in COP was 46.6%. This improvement in COP is very significant, as about 46% of compressor energy can be saved.

## 7. Conclusions

Indoor air quality has become an important public health issue worldwide. Past studies have shown that more fresh air supply could result in more energy use of air-conditioning. However, increasing energy use would cause adverse effects on global warming. The IDOAS proposed in this study might provide a part of the solution to the above problem. In comparison to the outdoor air unit that coupled to a central system, IDOAS was configured in a unique unitary system to integrate enthalpy recovery and a cooling system in a single housing. An experimental unit of IDAOS was designed and tested. The test results showed that, after enthalpy exchange, outdoor air could be brought to a lower temperature and lower humidity, with the advantages of reducing the energy needed to condition the outdoor air. Moreover, a lower temperature and lower humidity of outdoor air could be supplied to the spaces for better thermal comfort. The refrigerant condenser was cooled by lower temperature of exhaust air. It resulted in higher cooling energy efficiency. The test results indicate that about 44% of cooling can be saved due to enthalpy exchange. The cooling efficiency in terms of COP was improved by about 46%. This novel unitary IDOAS could be installed at the point of use and operated independently to the central air-conditioning system. Therefore, the application of IDOAS could save tremendous piping costs and energy needed to deliver chilled water compared to a conventional outdoor air unit. Saving energy and resources is becoming more important than ever as global warming is more severe as days pass. The test results obtained for the proposed IDOAS were encouraging and are worth further research and development.

**Author Contributions:** Conceptualization, Y.K.C.; methodology, Y.K.C.; formal analysis, Y.K.C. and J.J.Y.; investigation, Y.K.C. and J.J.Y.; resources, Y.K.C.; writing—original draft preparation, J.J.Y.; writing—review and editing, Y.K.C.; project administration, Y.K.C.; funding acquisition, Y.K.C. All authors have read and agreed to the published version of the manuscript.

**Funding:** This research was funded by Ministry of Science and Technology, grant number MOST 108-2221-E-027-042.

**Acknowledgments:** This research was funded by the Ministry of Science and Technology of Taiwan by research grant MOST 108-2221-E-027-042, it is acknowledged hereby gratefully.

**Conflicts of Interest:** The authors declare no conflict of interest

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
