# Peer review of "A Integrated Dedicated Outdoor Air System to Optimize Energy Saving"

_sustainability, doi:10.3390/su12031051_

Round 1

Reviewer 1 Report

        The authors measured the energy performance of a special and commercial HVAC system, called as DOAS (Dedicated Outdoor Air System).  The following comments are seriously addressed to turn down its publication in the Sustainability.

In order to understand the impacts on indoor air quality (IAQ) using DOAS in the present study, the authors should provide a table regarding the concentrations of indoor air pollutants, which include particulate matter (PM10, PM2.5), SO2, NO2 and O3 based on the IAQ standards in Taiwan. This paper failed to provide novel research on the heat transfer or heat balance theory. Also, the DOAS was a commercial product or system. The descriptions about the analysis of temperature, relative humidity, air velocity (or air flow rate) and power supply, and the heating/cooling equipment (i.e., compressor, evaporator, condenser and heat exchanger) were too simple. Please present their detailed information (e.g., manufacturer or model No.). The authors should discuss the present results (i.e., COP, energy saving percentage) with similar studies regarding the use and the DOAS. The experimental data lacked repeatability or reproducibility test. There are also many minor grammar mistakes that should be addressed and checked:

        L26  “been recognized that sufficient fresh outdo”

        L27  “ or air ventilation is imperative ……”

Author Response

First of all we would like to thank the comments from the reviewer. The manuscript has been revised according the comments given.

Pollutants concentration standard of Indoor Air Quality Act is added in Table 1. IDOAS built and tested in this study was not a commercial product. The total heat exchanger was made by Seibu Giken in Japan and the compressor was made by Rechi in Taiwan. The specifications are given in Table 2 (Table 3 in the revised version). The manufacturers are not mentioned to avoid commercialization. The heat exchangers (condenser and evaporator) with rated capacities are also described in Table 2 (Table 3 in the revised version). The specifications of the total heat exchanger is also given in Table 2 ((Table 3 in the revised version). The authors made careful measurement of the airflow rates of exhaust and intake that are important to the accuracy of the performance. It can be seen in Figure 6 the measurements to determine the states of the air so to determine the performance of IDOAS. Enthalpy and heat exchange efficiencies for exhaust and outdoor air intake given in Equations (4~9) indicate the good agreement of energy balance. The authors would like to explain IDOAS is novel as it integrates cooling and enthalpy exchange in a configuration that can achieve optimized energy saving explained with Figure 4. The primary objective is to prove that integrating cooling and enthalpy exchange has advantages of energy saving while maintain good air quality. It has been explained with Figure 4 that lower temperature and lower humidity outdoor air can be supplied to the room with IDOAS. Some references are added, The writing has been revised.

We have made good effort to design, fabricate and test the IDOAS. We believe that the results given in manuscript are of values and will be interesting to many researchers.

Reviewer 2 Report

Discuss in the intro paragraph how indoor quality immediate and long-term effects on health. Then move to the different kinds of pollutant sources (e.g. smoke, fuel burning, building materials, etc.). Talk in more detail in the introduction about the connection between health, energy efficiency, and climate change.

The paper moves from introduction to the advantage of dedicated ventilation unit. The intro offer a limited lit review, but the paper would benefit from a lit review that emphasizes public health, climate change and energy choices.

Not much is said about the weather in Taiwan and how the energy savings would be different in other geographies. How the results would change by location? This is a major limitation that should be addressed.

The conclusion is very thin. Bring back the literature and previous studies. What is different and new. Talk here about the limitations, emphasize here the site-specific information. Discuss the potentiality of these findings for benefiting overall public health and ameliorating the effects of climate change.

Author Response

We appreciate the comments from the reviewer. The manuscript has been revised according to the comments given.

Some references have been added. Pollutants concentration standard of Taiwan Indoor Air Quality Act has been added in Table 1. Indoor air quality as a concern of public health has been discussed with reference to a highly referred literature [2]. The problem of more outdoor and higher energy use has been discussed in full details with the addition of two references [8, 9]. The description of the climate of Taiwan has been added in the paragraph after Figure 5. Climatic zone of Taiwan has been added and with description of summer peak temperature. The issue of climate change and warmer outdoor air has been mentioned specifically with a discussion of the advantage of latent heat exchange in humid region and sensible heat for higher latitude countries, in the paragraph after Equation (9). The conclusion has been revised extensively to include the principles of IDOAS and with discussion of energy saving.

Round 2

Reviewer 1 Report

The authors failed to submit the responses of Point 1 and Point 4.  For example, it should show the data on the concentrations of IAPs, not the standards of IAPs in Taiwan.

Author Response

The manuscript has been revised extensively to respond to the reviewer comments. Fresh outdoor air can dilute the pollutant concentration and maintain good air quality. It is a design procedure described in ASHRAE standard 62.1, Ventilation for acceptable air quality [ref. 17]. The IDOAS proposed is a dedicated unit for fresh air ventilation. The authors are well aware of commercial air cleaning products such as using ultra violet and TiO2, and some other specific application such as chemisorption. It is not in the scope of this study. Principles of enthalpy exchange have been elaborated further in the paragraph after Equation (1). Equation (2) has been added for further clarification. Experimental tests have been described in details. IDOAS was designed by the authors, more description of IDOAS specification has been added. The conclusion has been revised.

Reviewer 2 Report

The article needs some proofreading, there are a lot of minor mistakes. 

In the introduction, I would like to see how this study is significant on a global scale, why this is important for other countries besides Taiwan. I think you are alluding to the importance, but maybe you can cite studies in other countries. 

The conclusion needs to go back to the initial literature presented and offer an analysis of what is different than other results or studies, what is new, and what is the significance of the current study. A conclusion is a good place for the limitations of the study and which other and future studies are needed. Right now the conclusion is very weak. 

Author Response

The manuscript has been revised extensively to respond to the reviewer comments. More description of the importance of IAQ on a global scale has been added in the introduction. References of regulation on IAQ and IAQ Certification Scheme in other countries or regions have added and described in the introduction. The conclusion has been revised. The improvements of IDOAS in comparison to current practice has been described. More elaboration has been given on global impact.

Round 3

Reviewer 1 Report

English language and style are fine/minor spell check required.

Author Response

Thanks for the review comments. I have thoroughtly revised the entire manuscript to make it more readable. I have also further elaborated on the background of the outdoor air system and basis of enthalpy exchange.